# Recent Progress of Using Knowledge Graph for Cybersecurity

Kai Liu [1], Fei Wang [1], Zhaoyun Ding [1],*, Sheng Liang [2], Zhengfei Yu [1] and Yun Zhou [1]

1   Science and Technology on Information Systems Engineering Laboratory, National University of Defense Technology, Changsha 410073, China; liukai18@nudt.edu.cn (K.L.); wangfei@nudt.edu.cn (F.W.); yuzhengfei19@nudt.edu.cn (Z.Y.); zhouyun@nudt.edu.cn (Y.Z.)
2   Center for Information and Language Processing (CIS), University of Munich (LMU), 80538 Munich, Germany; shengliang@cis.lmu.de
*   Correspondence: zyding@nudt.edu.cn

**Abstract:** In today's dynamic complex cyber environments, Cyber Threat Intelligence (CTI) and the risk of cyberattacks are both increasing. This means that organizations need to have a strong understanding of both their internal CTI and their external CTI. The potential for cybersecurity knowledge graphs is evident in their ability to aggregate and represent knowledge about cyber threats, as well as their ability to manage and reason with that knowledge. While most existing research has focused on how to create a full knowledge graph, how to utilize the knowledge graph to tackle real-world industrial difficulties in cyberattack and defense situations is still unclear. In this article, we give a quick overview of the cybersecurity knowledge graph's core concepts, schema, and building methodologies. We also give a relevant dataset review and open-source frameworks on the information extraction and knowledge creation job to aid future studies on cybersecurity knowledge graphs. We perform a comparative assessment of the many works that expound on the recent advances in the application scenarios of cybersecurity knowledge graph in the majority of this paper. In addition, a new comprehensive classification system is developed to define the linked works from 9 core categories and 18 subcategories. Finally, based on the analyses of existing research issues, we have a detailed overview of various possible research directions.

**Keywords:** cybersecurity knowledge graph; construction technology; application scenarios; cyberthreat intelligence extraction; cyberattack analysis; cyber threat prediction



## 1. Introduction

The scale of the cyberspace is gradually expanding from the traditional Internet to various areas such as manufacturing, agriculture, aviation, healthcare, and so on, as new information technologies and applications are developed. As a result, cyberspace can comprise interactions between industrial physical systems, human social systems, and network information systems and has become an increasingly complex infrastructure for social development. The opportunities left for attackers are increasing. Because of their combination of cyber and physical assets, the consequences of cyberattacks are becoming increasingly serious. The cyberattack experienced by Colonial Pipeline is an example of how a cyberattack can impact the physical world. The cyberattack shut down a pipeline that supplies 45% of the East Coast's fuel, leading to a USD 5 million economic loss in fuel delivery disruption and panic buying across the United States [1]. Given the growing number and severity of attacks and malware, the scarcity of qualified cybersecurity personnel is a cause for concern [2]. Since neither the number of available people nor the required skills can be increased overnight, companies must increase the development of technologies for modeling experts' knowledge and experience. The integration of automation, intelligent technology, and attack defense technology has become one of the inevitable trends in the development of cybersecurity technology.

Cyberattacks and counterattacks take place in dynamic, complex environments, with a variety of factors influencing attack success and mission impact. The cyberspace environments are continually changing, with the applications installed, machines added and removed, etc., which is one of the main obstacles [3]. On the other hand, the information asymmetry between the offensive and defensive sides in cyberspace is becoming more and more obvious [4]. For example, when confronted with a constantly updated new vulnerability or attack pattern, defenders often feel helpless to grasp the most up-to-date attacking techniques and vulnerability information, as well as the corresponding effective defense strategy, to maintain a balance with the attacker. The long persistence and highly concealed characteristics of modern attacks, such as advanced persistent threat (APT) attacks [5], make the limitations of traditional defense technologies based on expert rules, machine learning, and deep learning become increasingly apparent. Relatively simple tasks, such as feature extraction [6], anomaly detection [7], and data classification [8], can no longer restore the full picture of attack behavior. Expert knowledge hidden in cybersecurity data is still a very important breakthrough in solving the above problems.

However, the amount of cybersecurity-relevant data generated in cyberspace has skyrocketed. The data are varied, fragmented, and heterogeneous, making it hard for cybersecurity managers to locate the information they need quickly [9]. Therefore, the current issue in cybersecurity analysis is not a shortage of available data but rather how to combine non-homogenous information from various sources into a single model in order to better understand the cybersecurity situation as well as provide auxiliary information for decision-making. The current emphasis of cybersecurity research is to extract correlations as well as potential attacks from cyber threat intelligence information. Technologies such as correlation analysis [10], causal inference [11], and semantic reasoning [12] technologies based on knowledge modeling have evolved into novel solutions in the context of big data.

A cybersecurity knowledge graph (CSKG), as a specific knowledge graph (KG) in the security area, is made up of nodes and edges that constitute a large-scale security semantic network, providing an intuitive modeling method for various attacks and defense scenarios in the real security world. The entities or abstract concepts (e.g., vulnerability name, attack pattern, product name, vendor) could be represented by nodes. The attributes or the relationships among entities are represented by edges. The nodes and the edges together form a KG. The advantages of the KG can be discussed under three aspects: first, utilizing KG construction and refining techniques, including ontology [13], information extraction (IE) [14,15], and entity disambiguation [16], KGs effectively extract and integrate existing knowledge from multi-source heterogeneous data. Second, it can describe cybersecurity knowledge structurally and relationally, as well as visualize it in a graphical format, which is very intuitive and efficient. Third, using semantic modeling, query, and reasoning technologies, a cybersecurity KG can imitate the thinking process of security specialists that aim to derive new knowledge (as known as new relations) or check data consistency based upon the known facts (i.e., triples) and logic principles [17]. Although the construction of CSKG recently caught the attention of both researchers and companies. Many kinds of CSKGs have been constructed from different perspectives of cybersecurity. While the majority of research has concentrated on how to create a complete KG, it is still inexplicit how to implement the KG to tackle real-world industrial difficulties in cyberattack and defensive situations. Some research teams have proposed some schemas and made some attempts, but there is still a long way to go before they can be put into practice. How, for example, should KGs be used in a specific network environment with specific network assets? Security managers in many organizations wonder whether the existing CSKG can be reused in their work and whether it fits into their existing IT infrastructure. Furthermore, few earlier papers addressed what kind of new knowledge the CSKG may infer in addition to new relationships.

Previous systematic reviews or meta-analyses of the CSKG have been mainly undertaken in its data processing, construction, and visualization. Zhang et al. [18] reviewed the literature about the application of KG only from the aspects of situation awareness,

security assessment and analysis, and association analysis, which was limited to the security assessment region. Noel [19] summarized the graph-based approaches for evaluating and enhancing network security in two major aspects: the "when" aspect and the "where" aspect. The first dimension covers three particular phases (prevention phase, detection phase, and reaction phase) of the security process. In the second dimension, it gave an expectation that incorporates various operational components (i.e., network infrastructure, cybersecurity posture, cybersecurity threats, and mission dependencies) into a unified knowledge base for many cybersecurity tasks. The other articles reviewed the research of CSKG mainly from the dimensions of data sources, ontology design [20], construction technologies [21], and reasoning methods [22]. As a part of the review, Ding et al. [23] briefly attempted to illustrate several application directions of CSKG based on the introduction of CSKG construction technologies. However, none of the survey papers mentioned above focused on the challenging problem of how to utilize the CSKGs to solve practical issues.

The goal of this paper is to motivate and give an introduction to the application scenario of KGs in cybersecurity. To provide a comprehensive survey of the existing literature, this paper first gives a summary of the background and construction methods of the CSKG. We applied a relevant set of keywords: cybersecurity knowledge graph, cybersecurity knowledge representation, cybersecurity ontology, threat intelligence extraction, cybersecurity information extraction, cybersecurity knowledge graph application, graph-based analytics, and association analysis. These keywords are restricted to the title, keywords, and abstract search archives published between 2004 and 2022. There were 113 publications in total found after the database search. The primary content of this paper focuses on providing an overview of existing application scenarios of CSKGs and related datasets found in practice. At the end, we discuss the research field's future directions.

Our main works are as follows:

- **A comprehensive review of existing application scenarios of CSKG**. We propose a novel classification framework for conducting a comprehensive review of the application scenarios of CSKG based on an investigation of the background and construction technology of CSKG.
- **We summarize the relevant datasets**. To facilitate CSKGs' future research, we provide a review of datasets and the analysis of open-source libraries for two tasks: the CSKG construction task and the task of information extraction.
- **Future directions**. This survey summarizes each category and suggests possible future study directions.

The rest of this paper is organized as follows. In Section 2, an overview of the construction methods for CSKG, including definitions, the building flow, ontology, named entity recognition methods, and relationship extraction approaches, is given. The usual datasets, as well as their inadequacies, are presented in Section 3 to aid in the application of CSKG and the extraction of information. In Section 4, an overview of the application progress of the KGs in the cybersecurity domain is given. In Section 5, we discuss the shortcomings of existing research before prospecting future research opportunities. Finally, we conclude this paper in Section 6.

## 2. A Brief Overview of Construction Methods for Cybersecurity Knowledge Graph

In the domains of logic and artificial intelligence (AI), the representation of knowledge has experienced a lengthy history of development. With the resource description framework (RDF) [24] and the web ontology language (OWL) [25] being released sequentially, the concept of modern KG has become popular since its was first introduced by the Google search engine [26] to improve the search engines' capabilities and the users' search quality. This section will provide a quick overview of the CSKG construction process from the following aspects: First, we present a framework for building a CSKG. Then, the security ontology design is described as representing knowledge in the security domain. Next, our review goes to tasks of named entity recognition. Finally, we discuss relationship extraction and other related efforts in this field.

### 2.1. Some Definitions

KGs are structured semantic knowledge bases that are used to symbolically describe concepts and their interrelationships in the physical world [27]. Formally, a KG can be typically defined as $G = (E, R, T)$, where $G$ is a labeled and directed multi-graph, and $E = \{e_1, e_2, \cdots, e_{|E|}\}$ and $R = \{r_1, r_2, \cdots, r_{|R|}\}$ are the set of entities and relationships, respectively. $|E|$ and $|R|$ represent the number of elements in sets E and R, respectively. Each triple is formalized as $T = \{(e, r, e') \mid e, e' \in E, r \in R\}$, which represents a fact of the relationship $\boldsymbol{r}$ from head entity $\boldsymbol{e}$ to tail entity $e'$. In the knowledge base, the triplets < entity, relationship, entity > and < concept, attribute, value > are the basic forms of $T$. As the essential components of the KG, the entities mainly include categories, object types, collections, and categories of things (e.g., product, vendor, vulnerability, attacker). Relationships link entities to shape a graphic structure, and attributes include parameters and characteristics such as Windows, Google.com, and so on.

### 2.2. The Building Flow of KG

Similar to the general KG construction process, the CSKG follows the process and framework of the general KG construction. Due to the relatively mature and complete knowledge data of this field, a top-down construction method [28] can be adopted for building CSKG. The fragmented domain data could be integrated under the guidance of a certain framework or a pre-designed cybersecurity ontology from domain experts. Then information extraction and entity alignment technologies can separate entities and relationships from the original cybersecurity data. The knowledge-reasoning technology can generate new knowledge based on the existing KG to provide support for prediction and inference tasks. The construction framework of the CSKG is shown in Figure 1.

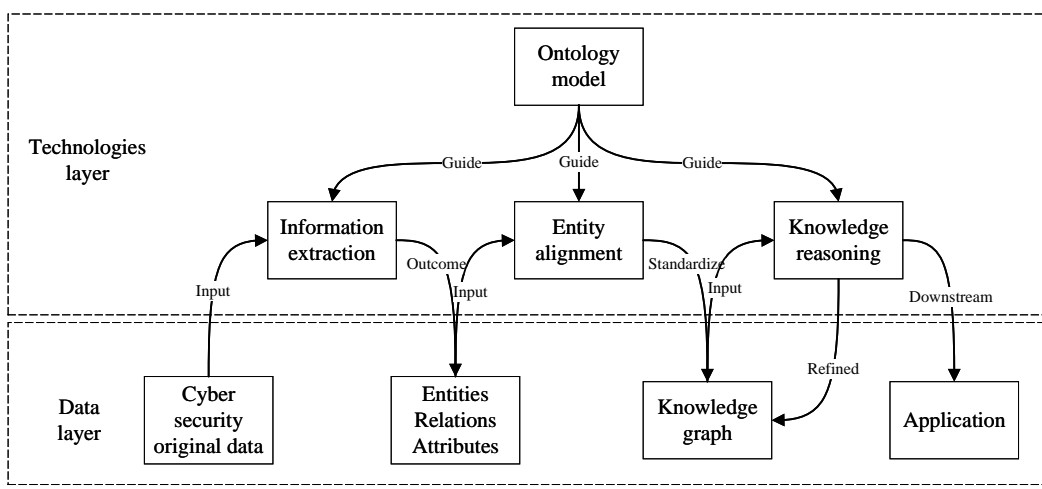

**Figure 1.** The construction framework of the cybersecurity knowledge graph.

### 2.3. Cybersecurity Ontology

The ontology of cybersecurity is being utilized to describe cybersecurity concepts and the relationships between them in a cybersecurity domain or even a broader range. These relationships and concepts have a widely accepted and unambiguous definition on which everybody in the shared range agrees, allowing humans and machines to interact [29]. Unified ontologies, such as STUCCO [30] and Unified Cybersecurity Ontology (UCO) [31], were created in the area of cybersecurity to incorporate heterogeneous data and knowledge schemas from diverse cybersecurity systems as well as the most widely used standards for the information exchange and sharing. Researchers have created various ontologies for various specific application scenarios, such as intrusion detection [32], malware categorization [33] and behavior modeling [34], Cyber Threat Intelligence (CTI) analysis [35], cyberattack analysis [36,37], threat and security evaluation [38,39], vulnerability analy-

sis [40], threat actor analysis [41], etc., as shown in Figure 2. Building a generic network security ontology in today's complex cyber environment is a difficult and time-consuming process that heavily relies on the domain knowledge and information technology knowledge of network security professionals. As a result, application scenarios should guide the design of appropriate security ontology. At the same time, dynamic and automatic enrichment of the information security ontology is required [42].

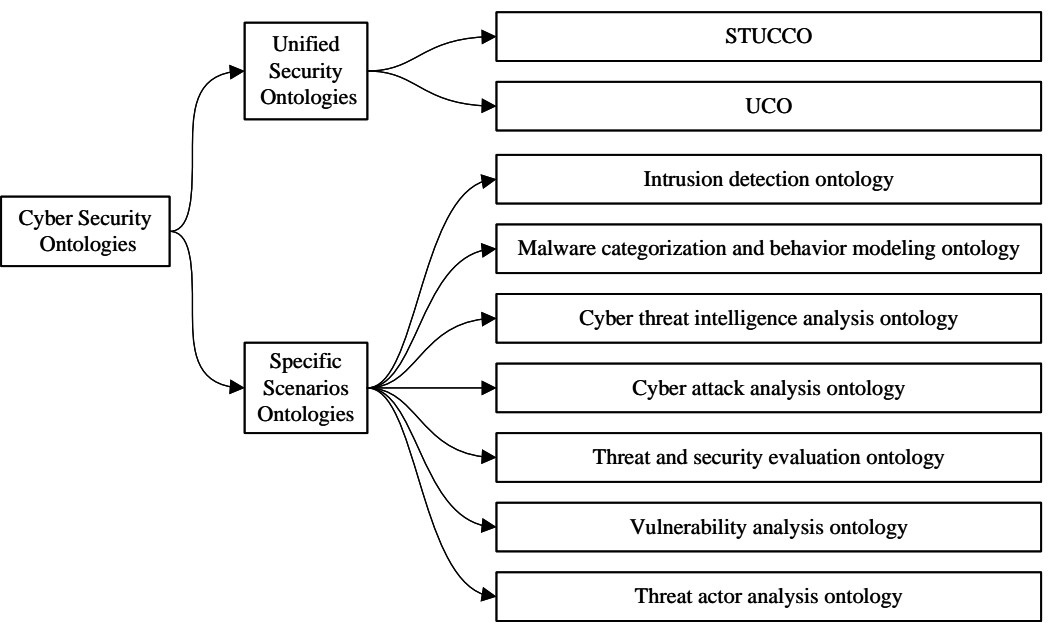

**Figure 2.** Cybersecurity ontologies.

### 2.4. Cybersecurity Entities Extraction

Information Extraction (IE) technologies have drawn incremental attention. At the moment, the two major tasks of IE are the Named Entity Recognition (NER) task and the Relation Extraction (RE) task. Traditional NER techniques could be divided into three main streams: the rule-based methods, the unsupervised learning models, and the feature-based supervised learning methods [43]. Rule-based methods, such as regular expression [44], bootstrap methods [45], etc., work well when the lexicon is exhaustive but cannot be transferred to other domains. Traditional statistical-based extraction methods [46], including the Hidden Markov model (HMM), the Decision Trees, the Maximum Entropy Model (MEM), the Support Vector Machines (SVM), and the Conditional Random Fields (CRF), achieve good results in comparison. They do, however, rely heavily on feature engineering, which has some drawbacks [47]. Deep learning outperforms traditional approaches in terms of representation learning as well as semantic composition, which are enabled by both vector representation as well as neural processing. Illustrated in Figure 3 are the three basic components (i.e., distributed representation, feature extractor, and decoder) and some corresponding instances of the deep learning NER approach.

This enables a computer to be fed raw data and uncover latent representations as well as processing required for categorization or detection automatically. At present, many methods [48–52] have been tried to be combined with deep learning, mainly including the transfer learning methods, multi-task learning, active learning, reinforcement learning, adversarial learning, the attention mechanism, etc.

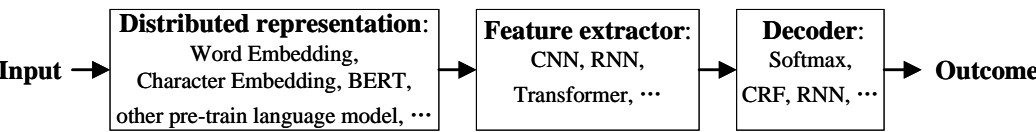

**Figure 3.** The deep learning approach for the NER flowchart.

*2.5. Relationships Extraction of Cybersecurity Entities*

The relationship between entities is an indispensable part of the KG. Abundant relationships weave independent entities together into a KG. Relationship extraction from unstructured text is one of the core tasks of the KG construction. To overcome the limitation of traditional methods, which highly rely upon the quality of manually created characteristics, Zeng et al. [53] proposed an end-to-end CNN-based technique that could collect key lexical and features at the sentence level automatically. Methods based on RNN or LSTM have been proposed one after another [54,55]. However, most supervised RE methods necessitate a substantial amount of training data with labels, which is costly to create. Distant Supervision (DS) [56] aids in the automatic dataset creation. Furthermore, multi-instance learning and sentence-level attention mechanism [57,58] were leveraged in the task of RE to reduce the noise introduced by DS. The abovementioned work typically solves IE tasks using the extract-then-classify or unified labeling methods. However, when it comes to extracting entities and relationships, these strategies either disregard the vital inner structure or have redundant entity pairs. To address these limitations, Fu et al. [59] presented a method of jointing the task of extraction of entities and relations, which outperformed the previous pipelined approach. Some researchers attempt to extract cybersecurity entity-relationship triples using a joint extraction model based on their unpublished datasets, which does not address the issue of labeled data scarcity [60,61].

However, there are three main challenges during extracting information from unstructured cybersecurity text. First, most previous IE research has concentrated on typical life events, such as those outlined by ACE [62] or the TAC Knowledge Base Population [63]. The needed domain-relevant expertise is one key difference between extracting life knowledge and cybersecurity knowledge. As a result, the IE task is hampered by a scarcity of large labeled training data. The intrinsic complexity of cybersecurity information is a second difference. A cyberattack event can be defined as an attack pattern comprising several activities that are either attempted or executed. Each mention of one of these acts can be regarded as a separate cybersecurity event description, increasing the possible choices for a cybersecurity event reference. Thirdly, there is much implicit information in the unstructured data that cannot be expressed explicitly.

To assure the quality of the KG, it is required to analyze and verify the knowledge, remove redundant knowledge, and resolve conflicting information to prevent errors from propagating in the reasoning process. Technologies, such as named entity linking and entity disambiguation [64], were proposed to refine the KG. Additionally, the KG, which is generated based on IE technologies, primarily contains relationships that are explicitly represented in sentences. It is also required to mine possible implicit knowledge and enrich the CSKG through reasoning [65]. Knowledge reasoning may be combined with specific task requirements, and it can employ association queries, reasoning with rule-based approaches, reasoning based on distributed representation, and the reasoning approaches with neural networks extensively [66].

## 3. The Datasets

Security analysts make decisions based on a wealth of knowledge to secure systems, including known and newly discovered threats, weaknesses, vulnerabilities, and attack patterns. Such knowledge is collected, published, and structured by research institutions, government agencies, and industry experts, e.g., the Computer Emergency Response Teams (CERTs) and MITRE [67]. The widely used standards include the vulnerabilities as well as the associated data published by the National Vulnerability Database (NVD) [68], such as CVE, CVSS, CWE, and CPE, and the prospective attacker exploits published by Common Attack Pattern Enumeration and Classification (CAPEC) [69]. In this section, we review the significant datasets for building CSKG by the categories as follows: (1) the datasets of open-source CSKG; (2) the datasets for IE in the cybersecurity domain; (3) other datasets that may inspire future researchers to come up with new solutions.

### 3.1. The Datasets of Open-Source CSKG

A comparative study of the datasets for CSKG is presented in Table 1. For each dataset, the purpose, data sources, and relevant papers (if available) are considered. These open-source CSKG are developed for different purposes, which means they need various data sources. For instance, the CWE-KG [70] was constructed to discover potential threats from Twitter data, so it needs information from CWE, CAPEC, and Twitter data. There are, however, obvious drawbacks to these datasets. One of these is that four of them do not give the statement document, e.g., paper, report. Moreover, the vulnerability KG [71] even only displays the visualization results on a webpage, without the statement of its construction method and experimental performance. The Open-CyKG [72] presents a CTI-KG framework that could extract vital cyber-threat knowledge from unstructured APT reports utilizing an attention-based neural Open IE model. Moreover, the MalKG [73] aims to integrate information on malware threat intelligence. The SEPSES CKB [74] provides the details of the CSKG dataset and the corresponding published paper [75]. While the knowledge from MITRE provided strong support for building this KG, this study neglects to include the Open-Source Cyber Threat Intelligence (OSCTI). Furthermore, these works are also limited by their only consideration of one language. Thereby, the researchers in the industry are beginning to look forward to the development of CSKG based on abundant threat intelligence and more languages.

**Table 1.** Comparative table of the datasets of open-source CSKG.

| Name and Year | Purpose | Data sources | Ref. |
|---|---|---|---|
| SEPSES CKB [74], 2019 | SEPSES CSKG with detailed instance data | CVE, CWE, CAPEC, CPE, CVSS. | [75] |
| CWE-KG [70], 2019 | A CWE KG supporting Twitter data analysis | CWE, CAPEC, Twitter data | - |
| CSKG [76], 2020 | Cybersecurity KG | CVE, CWE, CAPEC | - |
| ICSKG [77], 2020 | KG for vulnerabilities of industrial control systems | CVE, CWE, CPE, CERT | - |
| Vulnerability KG [71], 2021 | Visualization web page of vulnerability KG | CVE, CWE | - |
| Open-CyKG [72], 2021 | An open cyber threat intelligence KG | APT reports, CTI reports | [78] |
| MalKG [79], 2021 | Open-source KG for malware threat intelligence | CVE, Malware reports | [73] |

### 3.2. The Datasets for Information Extraction Task

Information extraction technologies are essential for generating CSKG. For the purpose of training robust IE models, providing quality-assurance annotated datasets is a crucial task that cannot be bypassed. This chapter summarizes several datasets for enhancing the works for knowledge extraction. From the discussion above, it is apparent that the datasets should be divided into two categories: the datasets for the NER task and the datasets for the RE task. Frankly speaking, the research around this topic is more than these, but the others did not publish their datasets. The datasets listed in Table 2 are the only part new researchers could download and use. Despite the fact that most datasets are defined only for the NER task, Rastogi et al.'s [73] new malware dataset is the only one for both the RE and the NER tasks. The entity and relationship types are generally defined in a specific security ontology, which restricts their use outside of the chosen domain. Although they are collected and annotated based on various data sources, most of them are generated with English corpora. On the other hand, a lot of security knowledge is released in a variety of languages, often intermingled with English, necessitating the creation of multilanguage datasets.

**Table 2.** The open datasets of information extraction task for cyber security.

| Dataset and Year | Task | | Entity Types | Data Sources |
|---|---|---|---|---|
| | NER | RE | | |
| Lal [80], 2013 | ✓ | - | Software, Network_terms, Attack, File_name, Hardware, Other_technical_terms, NER_modifier | Blogs, Official Security, Bulletins, CVE |
| Bridges et al. [81], 2013 | ✓ | - | Vendor, Product, Version, Language, vulnerability, and vulnerability relevant term | NVD, OSVBD, Exploit DB |
| Lim et al. [82], 2017 | ✓ | - | Action, Subject, Object, Modifier | APT reports |
| Kim et al. [83], 2020 | ✓ | - | Malware, IP, Domain/URL, Hash, their categories | CTI reports |
| Rastogi et al. [73], 2021 | ✓ | ✓ | Malware, MalwareFamily, Attacker, AttackerGroup, ExploitTarget, Indicator, etc. | CVE, Malware reports |

### 3.3. Other Datasets for the Application of CSKG

Transferring theoretical framework to practice by mastery of the environment, understanding the actions of threat actors, integrating external intelligence, and stockpiling basic knowledge. In 2021, Zhang et al. [84] categorized the data demand from four dimensions: environment data (e.g., assets and their weakness), behavior data (such as network alerts, terminal alerts, and logs), threat intelligence (e.g., internal and external CTI), and knowledge data (such as ATT&CK and CAPEC). However, the current studies highlight the threat intelligence and the knowledge data in building CSKG and ignore the environment and behavior data of the target IT system. Furthermore, there is still no mature and unified specification to describe these data.

### 4. The Application Scenarios

Knowledge graph technology has sparked a lot of research interest in recent years, thanks to its introduction by Google. In the cybersecurity domain, research on KG can be divided into two categories, study of the KG construction techniques and research of applications. Construction technique studies concentrate on the information extraction technologies, knowledge representation methods, knowledge fusion technologies, and knowledge reasoning methods in graphs [85], such as accurately attaching entities and relationships to the KG after extracting them from the textual corpus and reasoning some new triplets out of such KG. However, research should emphasize the application of CSKG to solve practical problems under different networks. This work provides a systematic survey on the recent progress of using knowledge graphs for cybersecurity. According to our recent survey, the majority of articles devoted to applying CSKGs to specific tasks have placed their interests in different particular phrases of the overall security process (assessing the overall situation of the network, discovering potential threats, and investigating the ongoing or ending attack), which will be presented in Sections 4.1–4.3 of this article. This article presents several specific applications from Sections 4.4–4.8 that will help operators and managers with decision-making, operation, vulnerability management, malware attribution, and analysis in conjunction with the physical environment. Section 4.9 introduces other possibilities of the application for CSKG, such as social engineering attack analysis and fake cyber threat intelligence identification. A taxonomy of the CSKG application scenarios is given in Figure 4.

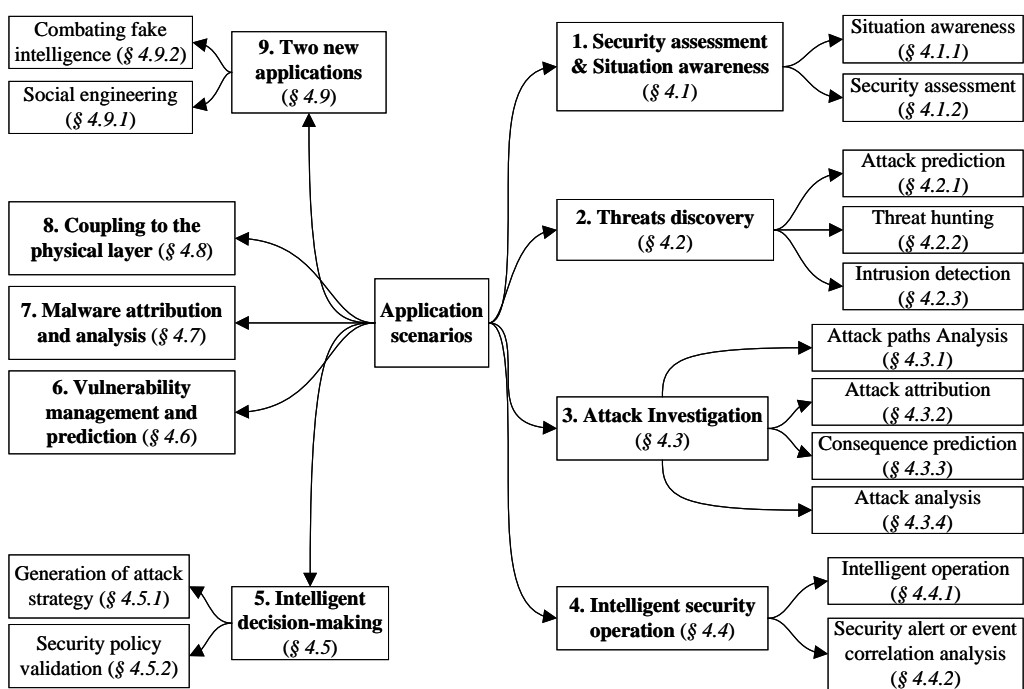

**Figure 4.** Application scenarios of cybersecurity knowledge graph.

### 4.1. Situation Awareness and Security Assessment

#### 4.1.1. Security Awareness

As the integration of equipment and services deployed on company networks has gotten increasingly intricate, assessing the overall security posture of an internal network as well as understanding its situation has become a demanding undertaking for human administrators. These multi-stage and multi-host attack scenarios must be addressed by an enterprise network security administrator. CSKG can play a vital role in situation awareness and security assessment. MITRE proposed a situation awareness system CyGraph [86], which is primarily oriented toward network warfare task analysis, visual analysis, as well as the management of cyber knowledge. CyGraph, as a KG with four layers: network infrastructure, security posture, cybersecurity threats, and mission readiness, aims to analyze attack paths, predict critical vulnerabilities, analyze the intrusion alarm, and analyze visual queries by bringing separated data and log events into an ongoing total picture. CyGraph provides some query-driven demonstration cases for stating its effectiveness but no corresponding datasets. Chen [87] combined an existing indicator system and situation detection model and proposed an attack situation detection scheme based on KG. It provides a novel feature for improving the accuracy of cybersecurity situation detection by abstracting the attack events (e.g., historical events, Internet news) as graph description. Wang [88] proposed a KG-based Network Security Situation Awareness model (KG-NSSA) to address the two classic problems: network attack scenario discovery and situation understanding. Different from the traditional alert-based attack scenario discovery approach, which is susceptible to a high level of redundancy and false positives, the scheme can effectively reflect the network attack scenario in the asset node situation with similarity estimation and the attribute graph mining method. Yi et al. [89] utilized the domain knowledge-based reasoning method to realize multi-source intelligence automatic correlation analysis and understand the satellite cyber situation.

4.1.2. Security Assessment

Table 3 shows a summary of the selected references about the knowledge graph's application in security assessment tasks. Wu et al. [90] proposed a novel ontology as well as a graph-based methodology for the task of security assessment. The ontology, which may be instantiated for specific networks, is designed to standardize the representation of security knowledge, such as assets, vulnerabilities, and attacks. Using the ontological model's inference capabilities, an efficient framework for generating attack graphs, identifying the possible attacks caused by published vulnerabilities, and assessing network security is proposed. The attack graph is a graphical representation and explanation of the attributes defined that are included in the method's final output. This study included an attack process flow diagram to clearly depict how an attacker might invade and compromise numerous goals of a test network across multiple hosts and multiple stages. This facilitates the enterprise security manager to complete security risk assessment tasks as well as respond to new threats. Inspired by the existing cybersecurity ontologies, such as STUCCO [30], UCO [31], and Cyber Intelligence Ontology [91], Kiesling et al. [75] proposed a publicly available CSKG with concrete instance information and illustrated its applicability for security assessment by two SPARQL [92] query example scenarios. It will query which data assets might be exposed in the local system model based on matching the information of organization-specific assets to a continually updated stream of existing vulnerabilities in order to estimate the possible effect of a newly discovered vulnerability. Pang et al. [93] proposed a security evaluation methodology of power IOT terminals based on KG with three dimensions: terminal assets, vulnerability, and intrusion alert, based on application scenarios as well as the power IOT terminal's threat characteristics. The approach performs a correlation analysis of the cybersecurity monitoring information of the IoT terminal with independent power and provides a terminal threat index that reflects the security condition of the terminal. This should have been a good attempt, unfortunately, the article did not give the details of the KG and method.

**Table 3.** The knowledge graph's application in security assessment task.

| Ref. and Year | Purpose | Method Description |
| --- | --- | --- |
| [90], 2017 | Generating attack graphs; Identifying the possible attacks; Assessing network security | Providing a novel cybersecurity ontology Providing a graph-based methodology for the purposes |
| [75], 2019 | Estimating the possible effect of new vulnerability | Constructing a CSKG SPARQL query method |
| [93], 2021 | Power IOT terminals' security assessment | Building a CSKG with three dimensions Correlation analysis of monitoring information |

*4.2. Threats Discovery*

Even with comprehensive monitoring, clever attackers may remain on a system for more than 100 days before being noticed, according to FireEye2018 [94]. Many issues, ranging from alert flooding to delayed response times, render conventional solutions useless and incapable of mitigating the harm caused by these attacks. The CSKG could fill this gap with the power of knowledge representation and reasoning. As listed in Table 4, this section presents the findings of the research about the application of CSKG, focusing on the three key themes, including attack prediction, threat hunting, and intrusion detection. Before the cyberattack, security analysts hope to spot signs of attacks ahead of time by attack prediction and threat hunting methods. During the attack, security administrators want to detect suspicious activities with intrusion detection technologies.

4.2.1. Attack Prediction

Narayanan et al. [95] developed a cognitive system that combines input from conventional sensors, dynamic internet textual sources, and KGs to identify cybersecurity

issues early. The author extended UCO so that it can reason over inputs from multiple network sensors, such as intrusion detection systems (IDS), Snort, and so on, as well as the knowledge from the cyber-kill chain. To express rules between entities, the Semantic Web Rule Language (SWRL) was utilized. The aggregator module was designed to combine alerts into a reasoning model. They proved its ability to identify newer attacks by putting it to the test against custom-built ransomware akin to WannaCry and displaying the timeline of the attack as well as the system's response activities. Unfortunately, this study solely described the system's architecture and did not include any additional information or data. Sun et al. [96] suggested a prediction approach of a 0-day attack route based on a cyber defense KG to address the challenge of attack prediction induced by the 0-day vulnerability. The KG was generated from three aspects (i.e., threat, assets, and vulnerability), which supported transforming the task of attack prediction into a KG link prediction problem. A path ranking algorithm was used to create the 0-day attack graph and discover the possible 0-day attack of the target system, according to the above methodology. The experimental results revealed that the suggested strategy might increase the accuracy of 0-day attack prediction with the aid of KG. Furthermore, employing the path ranking algorithm can aid in tracing the causes of predicted outcomes in order to increase the explanatory ability to forecast.

**Table 4.** The knowledge graph's application in threat discovery task.

| Tasks | Ref. and Year | Purpose | Method Description |
|---|---|---|---|
| Attack prediction | [95], 2018 | Identify cyber-attacks early | SWRL method<br>The UCO ontology extension<br>Combining multi-information into KG<br>Reasoning model |
| | [96], 2022 | Discover the possible 0-day attack | Building a cyber defense KG with 3 aspects<br>Path ranking algorithm |
| Threat hunting | [97], 2021 | Log-based cyber threat hunting with external OSCTI knowledge | Pipeline IE model for building threat behavior KG<br>A querying method based on system auditing<br>A threat behavior query language (TBQL)<br>A query synthesis technique |
| Intrusion detection | [75], 2019 | Estimating the possible effect of new vulnerability | A SPARQL query method<br>Link NIDS alerts to SEPSES KG |
| | [98], 2020 | Detecting the DDoS attacks TCP traffic | Expressing the connection between 2 hosts<br>Calculating the 1-way transmission propensity value<br>Finding the host of the initiator of a DDoS attack |
| | [99], 2021 | Detecting malicious traffic generated by DDoS attacks | DDoS attack malicious behavior KG creation<br>Behavior reasoning based on the KG |
| | [100], 2021 | Identifying unusual behaviors in industrial automation systems | Using a readily available ontology to build a KG<br>Combining 3 major sources of knowledge into KG<br>Machine learning abnormal detection based on KG |

### 4.2.2. Threat Hunting

For the task of cyber threat hunting, a system was developed by Gao et al. [97], aiming at facilitating log-based cyber threat hunting by leveraging vast external threat knowledge provided by OSCTI [101]. The system is composed of two subsystems, i.e., a pipeline IE model for building threat behavior KG and a querying subsystem based on system auditing, which can collect audit logging data across hosts. This study also includes a threat behavior query language (TBQL) and a query synthesis technique that automatically synthesizes a TBQL query based on the threat behavior KG with event sequence information, which could be used to discover matched system auditing records. Nevertheless, one of the limitations of this system is that it does not consider the attacks that are not detected by the system auditing. Furthermore, existing methods frequently have significant limitations in terms of the interpretability, quantity, and relevancy of the warnings issued.

### 4.2.3. Intrusion Detection

Besides the intrusion detection methods mentioned by [102], the CSKG could also be constructive in detecting intrusion. Kiesling et al. [75] gave a query-based case to demonstrate how network intrusion detection system (NIDS) alerts may be linked to the SEPSES CSKG to gain a better knowledge of possible threats and current assaults. Chen et al. [98] proposed a detection method of a DDoS attack based upon a domain KG, which is mainly aimed at DDoS attacks on TCP traffic. The KG is used to express the communication process of TCP traffic between two hosts. Together with calculating the value for one-way transmission propensity, a threshold was set to determine whether the source host is the initiator of a DDoS attack. To comprehensively describe the DDoS attack, Liu et al. [99] constructed a DDoS attack malicious behavior knowledge base, which contains two parts: a malicious traffic detection database and a network security knowledge base. The front one is responsible for detecting and classifying malicious traffic generated by DDoS attacks. The network security knowledge base, which includes the network topology graph, malicious behavior traceability graph, malicious behavior feature graph, and traffic behavior KG, is at the heart of the malicious behavior knowledge base of DDoS attacks. Data structure processing, malicious behavior KG creation, behavior reasoning, and feedback are all handled by the network security knowledge base. In 2021, Garrido et al. [100] applied a machine learning method to KGs to identify unusual behaviors in industrial automation systems integrating IT and OT elements. Using a readily available ontology [103], this study builds a KG by combining three major sources of knowledge: automation system information, application-level observations (e.g., data access events), and network observations (e.g., connections between hosts). Inspired by KG completion methods, this study adopts a graph embedding algorithm to rate the likelihood of triple assertions emerging from observed security events. Experimentally, the suggested method produces intuitively well-calibrated and interpretable alarms in a variety of contexts, pointing to the relational machine learning potential benefits on KG for the task of intrusion detection. Although the results are generated on a reduced-scale prototype and without the help of CTI, the present research explores, for the first time, the synergistic combination of KG and industrial control systems.

### 4.3. Attack Investigation

CSKG is fast becoming a key instrument in attack analysis. In the chapter that follows, we will discuss recent progress on how to apply KG to attack analysis from the following four aspects: attack path analysis, attack attribution, consequence prediction, and attack analysis. Table 5 gives the relevant references for the four concrete topics and introduces their purpose and methods.

**Table 5.** The knowledge graph's application in attack investigation task.

| Tasks | Ref. and Year | Purpose | Method Description |
|-------|---------------|---------|--------------------|
| Attack path analysis | [104], 2015 | Predicting possible attack paths | A unique model of attack graph<br>Predicting possible attack paths based on events<br>Correlating attack events with attack paths |
| | [105], 2019 | Generating an attack path and improving the assessment of vulnerability | Building cyber-attack KG with 4 types of entities<br>Determining penetration paths with graph method |
| | [37], 2019 | The penetration path generation | Providing a two-layer threat penetration graph<br>Description of attack-related resources of hosts<br>Generation of the penetration path |
| | [106], 2022 | Obtaining the maximum probability vulnerability path | Providing an expansion attack graph<br>Obtaining maximum probability vulnerability path<br>Providing the success rate of attacking and the loss |
| | [96], 2022 | Representing and generating 0-day attack paths | Approach of link prediction<br>Path ranking algorithm |

**Table 5.** *Cont.*

| Tasks | Ref. and Year | Purpose | Method Description |
|---|---|---|---|
| Attack attribution | [107], 2018 | Attack attribution | CSKG of space-ground information network<br>An automated cyberattack attributing framework<br>Attack attribution by querying the CSKG |
| | [108], 2020 | Attack attribution based on provenance graph construction | Provenance graph construction from 3 dimensions<br>Causal analysis based on provenance graph<br>Extending the graph from single host to multi-host |
| Consequence prediction | [109], 2018 | Predicting the links, CWE triples and threat consequences based on CWE KG | Vector embedding generation<br>Building a KG-based CWE data<br>3 different reasoning methods based on CSKG |
| | [110], 2020 | Attack consequence prediction | Building machine learning classification models<br>Providing dataset includes 93 different attacks |

### 4.3.1. Attack Path Analysis

As mentioned above (in Section 4.1), the CyGraph could query out potential attack paths based on the network environment. Similar to CyGraph, Neol et al. [104] illustrated a graph-based strategy with a unique model of attack graph that mixes a complicated mix of network data, such as the network topology, firewall strategies, vulnerability intelligence, attack patterns, and threat alerts, via cybersecurity data standardized languages. Furthermore, the author, created a model that predicts possible attack paths based on network events (intrusion alerts, sensor logs, etc.). Correlating observed attack events with prospective attack pathways provides the optimum reaction choices, particularly for safeguarding important assets, and enhances situational awareness, such as missed attacking steps inferring and false positive minimizing. The attack graph that results is stored in the Neo4j database [111] for query and visualization. Despite its efficiency of query and visualization for potential attack paths, the proposed KG still faces several disadvantages. First, this research did not show us how to use KG to infer new knowledge. Secondly, the corresponding datasets, such as the input format of alerts data and firewall policies, were not demonstrated clearly. Finally, the OSCTI (the knowledge provided by METRE) is not reflected in the architecture.

To extend the information on the attack path, Ye et al. [105] designed a cyber attack KG with four types of entities, including software, hardware, vulnerabilities, and attack entity. With the help of four kinds of attributions of attack entity (i.e., attack conditions, attack methods, success rate, and earnings), this research used KG to generate an attack path and improve the assessment of vulnerability rather than rely on the CVSS score [112]. Thanks to the knowledge representation and information management ability of KG, the attack path could update local information based on multiple sources. To further increase efficiency, a graph-based strategy for determining the ideal penetration path is proposed, taking into account insider and unknown attacks. Wang et al. [37] defined a two-layer threat penetration graph (TLTPG), where the upper layer is a penetration graph of the network environment, and the lower layer is a penetration graph between any two hosts. The KG was used to describe the attack-related resources (e.g., software, vulnerability, ports in use, and privilege of a successful attack) of each host, which would be of great benefit not only to generate the penetration path between hosts but also to integrate collected information of 0-day vulnerability attack for unknown attack prediction. In the power networks, Chen et al. [106] generated an expansion attack graph for obtaining the maximum probability vulnerability path and providing the success rate of attacking the power grid and the loss.

As was discussed in the "attack prediction" section, the KG could also be used to represent and generate 0-day attack paths with the approach of link prediction and path ranking algorithm [96]. It comprehensively considered the existence, availability, and impact of vulnerability, as well as the knowledge of attack intent and asset types.

### 4.3.2. Attack Attribution

As a defender, we must be able to answer questions such as who attacked me, where the attack point is, and what the attack path is in order to gain a competitive advantage in cyberwarfare. This step is known as attack attribution. The attack source, intermediate medium, and corresponding attack path can all be determined using attack attribution technologies, allowing for more tailored protection and countermeasure techniques to achieve active defense. As can be seen, the attribution of attacks is a crucial step in the transition from passive to active defense. Based on an ontology with six dimensions, namely host asset, vulnerability, attack threat, evidence, location, strategy, and the relationships between them, Zhu et al. [107] constructed a CSKG for the network of space-ground integration information. In addition, each dimension has several unique attributes and data sources. The research proposed an automated cyberattack attributing framework. Attack attribution could be performed from several aspects by querying the established CSKG. As an example, given in the article, based on the host asset level, security employees could query the KG to find out the vulnerable host asset suspected of being attacked, associated vulnerability, and attribution strategy in sequence. Then, by implementing the corresponding attribution strategy, the attacker's evidence and locations could be identified, and the attacked host asset can be determined. Xue et al. [108] analyzed the existing provenance graph construction technology based on causation in NSFOCUS blogs. The study introduced the provenance graph construction from three dimensions, including the terminal dimension, the perspective of Syslog and application log correlation, and the association of network and terminal. The terminal perspective method focused on the relationships between processes, files, and filenames in a single isolated host and ignored the application log, which was replenished by the second dimension. Moreover, the third level method extended the provenance graph from a single-host to a multi-host network, which could be further enhanced through causal analysis to gett a complete attack process. However, this study did not consider the semantic context and OSCTI provided by CSKG.

### 4.3.3. Consequence Prediction

Common software weaknesses, such as poor input validation and integer overflow, can directly or indirectly impair system security, resulting in negative effects such as denial-of-service (DOS) and unauthorized code execution. Understanding the consequence of weakness becomes significant to assessing the risk of a system and to take prompt response. In 2018, Han et al. [109] built a KG based on common weakness enumeration (CWE) [113], which provides detailed information regarding vulnerabilities such as the textual descriptions, relationships between software weaknesses, and common effects. The available CWE data do not allow sophisticated reasoning tasks, such as missing links prediction and the prediction of common consequences. This study developed a description-embodied, translation-based knowledge representation learning approach for embedding both the weaknesses and their relationships into a semantic vector space. Following the vector embedding generation, extensive experiments were conducted to estimate the performance of KG in knowledge acquisition and inference tasks. In this study, CSKG could be exploited for three different reasoning tasks: link prediction task based on CWE, CWE triple categorization task, and threat consequence prediction task. Datta et al. [110] transferred the consequence prediction problem to the classification task by introducing a dataset and building machine learning models and natural language processing (NLP) models. The cyberattack dataset includes 93 different assaults and their descriptions, which are annotated with technical and nontechnical consequences. The goal is to provide security researchers with tools that make it simpler to convey the consequences of an attack to diverse stakeholders who may have little to no cybersecurity experience. Furthermore, the suggested technique can lessen researchers' cognitive strain by automatically forecasting the consequences when new attacks are identified.

### 4.3.4. Attack Analysis

Besides the above application, Qi et al. [114] built a CSKG, which includes two subgraphs: CSKG and scene KG, for attack analysis. The CSKG is the core graph representing the knowledge about vulnerabilities, attacks, assets, and the relationships among them, which can be collected from various websites publishing vulnerability and attack analysis intelligence and can be updated gradually. Scene KG is an extended graph constructed using node and connectivity information of the network involved in a specific attack. The data gathering system and detection system provide the input data for the whole analytical framework. Composite attack chains are generated from several single attacks using the CSKG, attack rule base, and spatiotemporal property restrictions. With so many alerts to analyze, cyber investigators frequently suffer from alert fatigue, leading them to disregard a high number of alerts and overlook real attacks. It has been shown that distinct attacks, independent of the vulnerabilities exploited or payloads performed, may use identical abstract tactics. Alsaheel et al. [115] presented ATLAS, a methodology that generates an end-to-end attack scenario from ready-made audit logs using a causal graph. ATLAS employs a revolutionary mix of NLP, causality analysis, and machine learning approaches to construct a sequence-based framework that extracts essential patterns of attack and nonattack behaviors from such a causal graph. Given a security event, an attack symptom node in the causal graph is generated at the inference phase. ATLAS then builds a collection of possible sequences linked with the symptom node, employs the sequence-based framework to identify nodes that contribute to the attack, and combines the discovered attack nodes to generate an attack story.

6G-oriented network intelligence requires knowledge from both within and outside the network. Therefore, Wang et al. [116] proposed a method for building cyberattack KGs based on CWE as well as CAPEC [69], which is deployed in Neo4j (a graph database). This study only introduced two query-based application scenarios in detecting and responding to DDoS flood attacks and multi-stage attacks based on Neo4j's query and display functionality, rather than based on the reasoning function of KG. This study just focused on the analysis and application with CVE and CAPEC, which did not cover the complete knowledge of cybersecurity.

### 4.4. Intelligent Security Operation
#### 4.4.1. Intelligent Operation

An AI-driven security operations framework was presented by Zhang et al. [84]. In this study, CSKG could support dynamic query and aggregation analysis of security data, improving the integrity of security data operation analysis. The KG is a unified data view, which can support the realization of multi-level technical capabilities such as subsequent risk perception, causal cognition, and robust decision-making. Some challenges were discussed from the aspects of data, models, and semantic context. However, this research did not demonstrate the specific method.

The white paper [117] introduced the application scenarios of KG in the field of security operation from three aspects: attack profiling, attack path investigation, and response mitigation strategy recommendation, as well as the challenges of intelligent operation. As a white paper, it aims to sort out directional content, such as demand scenarios, application solutions, and technology prospects in this research field, but it will not involve technical details.

#### 4.4.2. Security Alert or Event Correlation Analysis

Given the ever-changing threat landscape, cybersecurity researchers overseeing the Security Operation Center (SOC) are frequently overburdened with various security events while also attempting to stay up with the most recent threats in the field. Effectively analyzing vast amounts of various alerts or event data brings opportunities to detect concerns before they become problems and to avoid further cyberattacks. Conventional techniques frequently store the many aspects of security information in distinct databases,

which results in the absence of synergies between the multiple dimensions. As illustrated by Xue [118], the main challenge faced by the application of cyber CSKG is that there is no direct connection between the KG based on abstract attack knowledge such as STIX 2.0 and the system and network logs that contain the behavior information. It is a semantic gap between them. For complicated cyberattacks, it is difficult to incorporate all context information quickly to initiate real-time and accurate analysis. To create the attack scene, traditional rule-based association analysis relies on expert knowledge, which lacks the capacity to reason automatically.

To address the aforementioned issue, Wang et al. [119] presented an integrated correlation analysis approach to a cybersecurity event. The approach included the vulnerability KG, threat intelligence KG, the network infrastructure KG, and intrusion alert KG into the CSKG, as well as documented the data sources for each dimension. Following alert normalization and alert fusion, the alert verification was conducted by judging whether the vulnerabilities of one alert are in the host vulnerability set. Furthermore, the attack thread correlation analysis process relies on the existing alerts to query the associated alerts, CVE items, and CAPEC items, which could be conducive to predicting the real purpose of attackers. In the authors' thesis [120], rebuilding the scene of a series of alerts based on KG was introduced in detail. The author conducts an experiment on the DARPA 2000 dataset to assess the performance of the proposed framework by comparing the number of remaining alerts after correlation analysis. This research showed an example of the use of KG for correlation analysis. Qi et al. [121] believed that cyberattacks involve various attack phases that are related to IDS alarms. Based on this thought, an association analysis model developed on cybersecurity attack events KG is presented to display a cyberattack scenario in a special air–ground integrated network graphically. The CSKG includes five tuples: attacks, alarms, events, relations, and the rules. The association analysis was used by calculating the coincidence degree between the gathered events sequence and the attacked events sequence in the KG.

However, due to the absence of a thorough knowledge of the integrated space–ground network as well as the limits of the present experimental settings, this article relied solely on simulation tests to validate the viability of the aforementioned approach. Manual analysis of logs often does not scale well and frequently results in a lack of knowledge and insufficient transparency about concerns. To address this issue, Ekelhart et al. [122] introduced a flexible framework for the automated construction of KGs from arbitrary raw log messages. The method closes a key gap and offers up a variety of data sources for KG construction by making the log data suitable for semantic analysis. As mentioned earlier in Section 4.2, Garrido et al. [100] proposed the application of machine learning on KGs to increase the utility of the IDS-generated alerts for human operators by improving their quality and relevance in modern industrial systems.

### 4.5. Intelligent Decision-Making

The present cybersecurity evaluation is also according to personal knowledge, and the intelligence level is insufficient. Enhancing the intelligence level of cybersecurity evaluation is an important topic that must be handled. It is necessary to investigate the decision model relevant to cybersecurity and increase the intelligence ability of cybersecurity assessment using KG technology. This section's goal is to describe numerous study scenarios of intelligent decision-making with KG, including attack strategy development and security policy validation.

#### 4.5.1. Generation of Attack Strategy

Analyzing the attack strategy from the attacker's point of view can assist in identifying current security issues and can give targeted protective recommendations. Compared with the query-based method of CyGraph, a cyberattack method recommendation model that relied upon KG was proposed by Ou et al. [123]. It contains a six-tuple KG construction schema based on four open databases (i.e., CVE, CWE, MSF, CAPEC), the collaborative

filtering recommendation that describes difference relationships between nodes by meta-path, a generator of recommendation list with calculating the correlation score of each path with node vector. In the second part, a recommendation algorithm for cyberattack entities is proposed by combining the method of machine learning feature extraction and the method of constructing a heterogeneous information network meta-path. Based on this KG, intelligent searching and recommendations of knowledge related to new threat intelligence can be achieved. Compared with the traditional content-based search recommendation method, this method is more accurate at predicting the weakness of vulnerabilities and can realize the prediction and recommendation of attack patterns based on the natural language description of vulnerabilities. Likewise, from the perspective of an attacker, Chen et al. [124] proposed a method for generating a knowledge-driven attack strategy in order to exploit various vulnerabilities in an industrial control network. The method consists of a vulnerability exploitation KG, an industrial control network graphical representation, and the reasoning rules rely on KG. It is a common idea among cybersecurity experts that look for attack paths at the device level depending on the attack process. The attack strategy formulation process can be partitioned into two steps: The first step is analyzing the current device-level nodes' several vulnerabilities and correlating them to the consequences and pre-conditions of exploitation. To devise a global attack strategy graphically, the research linked the device-level nodes depending on firewall access rules as well as other protection devices after formulating the sequence in which all device vulnerabilities are exploited. Currently, this proposed KG is applied to analyze numerous vulnerabilities on a small-scale industrial control network to generate attack paths. With the expansion of KG, such as supplementing with other threat intelligence, more and more attack strategies need to be generated, especially the most cost-effective attack strategy.

### 4.5.2. Security Policy Validation

Vassilev et al. [125] proposed a four-layer (i.e., ontological level, heuristic level, workflow level, and process-level) architecture for CTI analysis, logical analysis, and validation of security policies. The architecture has been verified using a collection of scenarios depicting digital banking's most frequent security risks, and a prototype of an event-driven engine for traversing intelligence graphs has been constructed. However, this framework was created particularly for use in digital banking and did not include any previously used datasets.

### 4.6. Vulnerability Management and Prediction

This section contains a number of case studies that demonstrate the different CSKG analytic ability in vulnerability management and prediction. Managing, detecting, measuring, and prioritizing vulnerabilities in a single system is a critical job and a prerequisite for threat mitigation and removal, and, therefore, for the successful protection of precious resources.

KG technologies provide an exciting opportunity to advance our knowledge of managing considerable vulnerability data by presenting them in a structured ontological format. Except for the use case in Section 4.2.3, another use scenario of the SEPSES KG [75] is a query-based example to state how the KG can support security analysts by generating the information organization-specific asset to a stream of known vulnerabilities that is constantly updated. Cybersecurity vulnerability ontology (CVO), a formal, conceptual knowledge representation model in the vulnerability management area, was created by Syed et al. [126]. Additionally, in this research, they utilized the CVO to develop a system of cyber intelligence alert (CIA) that sends out threat alerts regarding potential vulnerabilities and countermeasures. At the practical level, its components contain the social media intelligence extractor-tagger (SMIET), the vulnerability repository and mapper, CVO, RDF converter, the cyber intelligence ontology (CIO), and the engine of cyber alerts rules. Finally, this study gave the evaluation approaches, corresponding results, and examples in

practice. Based on the industrial internet security vulnerabilities, an industrial CSKG was built and stored in Neo4j by Tao et al. [127] in order to analyze, query, and visualize from the temporal, spatial, and correlation dimensions.

When confronted with actual intrusions, CyGraph correlates intrusion alerts to published vulnerability pathways and recommends the appropriate courses of action for reacting to attacks. CyGraph creates a predicted model of likely attack pathways and major vulnerabilities based on queries. As previously stated, by constructing a knowledge representation learning approach (translation-based, description-embodied), a CSKG based on CWE might be utilized to infer incomplete relationships and common effects. To find hidden relationships among weaknesses, Qin et al. [40] proposed a query-based model for analyzing and reasoning new knowledge automatically. The reasoning flow of the sample CWE Chain was demonstrated based on a vulnerability KG (VulKG), which covers the vulnerability data from NVD, CVE, CWE, and CPE. However, the example could merely partially take over the place of the analysis and labeling work of security specialists under some specific scenarios, where the operator needs to know the query target previously. For effectively managing the sparse or inaccurate malware threat information, a malware KG called MalKG was established by Rastogi et al. [73], which is the first open-source automated malware threat intelligence KG. Additionally, there are approximately 40 thousand triples in the provided MalKG dataset (i.e., MT40K), which include 27,354 unique entities and 34 relationships. The study also manually curated a benchmark KG dataset called MT3K, with 5741 unique entities and 22 relationships, forming 3027 triples. It demonstrated the prediction capabilities of MalKG using two use cases in predicting new information. One of the application scenarios is predicting and sorting all the potential vulnerabilities or CVEs of the malware-impacted software system by comprehensive utilization of information from the network environment, malware, and KG. A vulnerability exploitation KG was built by combining and extracting multi-dimensional domain knowledge, as has already been noted in [124]. Attack strategies depending on KG enhance the performance in comprehensive vulnerability exploitation and flexible response by analyzing each device-level node. Based on an industrial network example, the feasibility of the method was investigated. Similarly, in Wang's study [128], chain reasoning and confidence calculation were also used to support vulnerability detection and finding latent relationships between CWEs. At the end of this research, similarity matching based on a source code level graph is used for judging the similarity between the target node and the node in the vulnerability database, which provides new insights into vulnerability mining. Wang et al. [129] extended the relationships in the vulnerability KG by identifying the alternative vulnerability with similar consequences.

*4.7. Malware Attribution and Analysis*

This section aims to discuss how to utilize the KG to analyze and attribute the malware. Najafi et al. [130] devised MalRank, a graph-based malware rank inference model aimed to predict a node's maliciousness by its associations with the other entities in the KG, such as common IP ranges or dns servers. This essay presented a KG that builds global relationships among entities detected in proxy and IDS logs, enhanced with related CTI and open-source intelligence (OSINT). The authors formulate threat detection in the security information and event management (SIEM) environment as a large-scale graph inference problem. MalRank maintains a high detection rate, beating its predecessor, belief propagation, both in terms of accuracy and efficiency, according to a series of trials using real-world data acquired from a worldwide organization's SIEM. It was also demonstrated that this method is useful in detecting previously discovered hostile entities, including IP addresses and malicious domain names. Besides the application scenarios reported earlier, MalKG [131] could also be implemented in the malware attribution scenario [73]. For example, given a newly discovered malware attack on one system, the analyst needs to build a fingerprint of the malware's origin by assembling sufficient features, such as author, campaign, and others. The goal of MalKG is to automate the prediction of these

features associated with a given malware; for instance, the newly discovered malware may share similarities with a disclosed malware linked to a certain APT group. As reported in the white paper [117], the profiling and automatic attribution of APT attacker gangs can be realized through the extraction of key elements of threat intelligence and dynamic behavioral reasoning. The key solution lies in establishing a unified language to describe the behavior and characteristics of different APT organizations, as well as in building a knowledge base about APT organizations. However, the white paper did not disclose the details of the related research.

### 4.8. Connection to the Physical System

CSKG uses big data analysis and graph mining technology to deeply analyze the coupling relationship between the information layer and the physical layer in the modern industrial control system and to realize the intelligence of "decision making, risk prediction, accident analysis, attack identification" and other capabilities assisted and automated processing. This section attempts to provide a summary of the literature relating to how to combine the CSKG with a physical network environment.

To illustrate various cybersecurity analytic capabilities in the MITRE's situational awareness system, CyGraph [86,104], they presented a simple network architecture for the case study. The architecture shows the fundamental connectivity among hosts, firewalls, routers, and switches. The internal network is divided into three security domains (mission client workstations, DMZ, and the data center). The internal network is protected from the outside by the external network firewall, while the vital data-center servers are protected by the internal firewall. The KG was created using information from the topology of the network, firewall rules, and vulnerability scan findings. To verify the effectiveness of the proposed method, a typical internal network architecture model with six types of elements is introduced in this paper [105]. In this architecture, the firewall isolates the Internet from the intranet router. The FTP server, host1, and host2 are directly connected to the router. Host1 and host2 can access the FTP server. The database server is connected to the FTP server to receive and respond to requests from the FTP server. For the purpose of generating penetration paths, the essay [37] designed an illustrative network example. The network contains a host on the Internet, a DMZ area, and three subnets. There is a web server in the DMZ area. Subnet1 has two devices (i.e., one Pad and a host), which can be connected to the Internet. Subnet2 has two hosts and cannot connect to the Internet. Subnet3 includes three servers, including a print server, file server, and data server. The attacker is a host on the Internet. It also considered the potential connection between subnet1 and subnet2 via USB. Similar to the architecture above, an experimental network environment was designed in the paper [96]. In this study, the experimental network contains two subnets protected by two firewalls separately and one DMZ also with a firewall. The two subnets connect to the Internet through the DMZ. Each part of the network has different assets, such as an email server and a web server; in the DMZ, two hosts and a file server are part of subnet1, and an application server is connected to subnet2. Building a suitable experiment network environment could be beneficial to demonstrate the effect of approaches and reproduce the attack and defensive process.

Despite the various traditional network architectures, it is also important to research the security of industrial control systems with a suitable experimental network. In the research of [100], a hardware prototype was described for evaluation based on the architecture of current industrial systems merging IT and OT aspects. A Siemens S7-1500 PLC is used for the automation side, and it is linked to peripherals through an industrial network. A conveyor belt driving subsystem, a human–machine interface (HMI), an industrial camera, and a distributed I/O subsystem with modules interfacing with different sensors for object location and other measurements are among these peripherals. Through an OPC-UA server, the PLC provides the values recorded by these sensors and the system state information. Therefore, the PLC connects to two edge computing servers. Thereafter, the network with main traffic flows was also displayed in this paper. A KG-based security assessment

technique for power loT terminals is provided [93] in order to perceive and measure enormous power IoT terminals' security risks and threats in real-time. However, this article did not describe a suitable network for evaluation. As analyzed previously, Chen et al. [124] used the domain KG to produce attack strategies by analyzing several vulnerabilities in the industrial control system. The topology of the target network is composed of the Internet, two firewalls, one router, an enterprise network, and an industrial ethernet. One firewall is used to protect all assets of the local network. The other one is situated between the enterprise network and industrial ethernet. The route is between the first firewall and the enterprise network, followed by the second firewall and industrial ethernet. The assets of the enterprise network include a web server, admin host, and printer. Some peripherals, such as an HMI, a data server, a workstation, and three PLCs with different end-effector devices (e.g., valve, flowmeter), are connected to the industrial ethernet. The attacker is a certain host from the Internet, and the PLCs are the attack target.

Based on the above analysis and related research, this paper sorts out a general experimental network architecture (as shown in Figure 5) to demonstrate the effect of potential security investigation approaches. This general network mainly contains four parts, including DMZ, a subnet connected to the Internet via a router, a subnet connected to DMZ, and an industrial control network connected to DMZ. Each subnet is isolated by a firewall, and the attackers usually start their offensive action from the Internet. A researcher could utilize it to adapt to a complex network by modifying or adding some devices, extending the subnets, or changing the connection mechanism. The network topology expresses the network environment. In addition, it should also include the software and hardware installed on each node, security protection measures, and existing vulnerabilities.

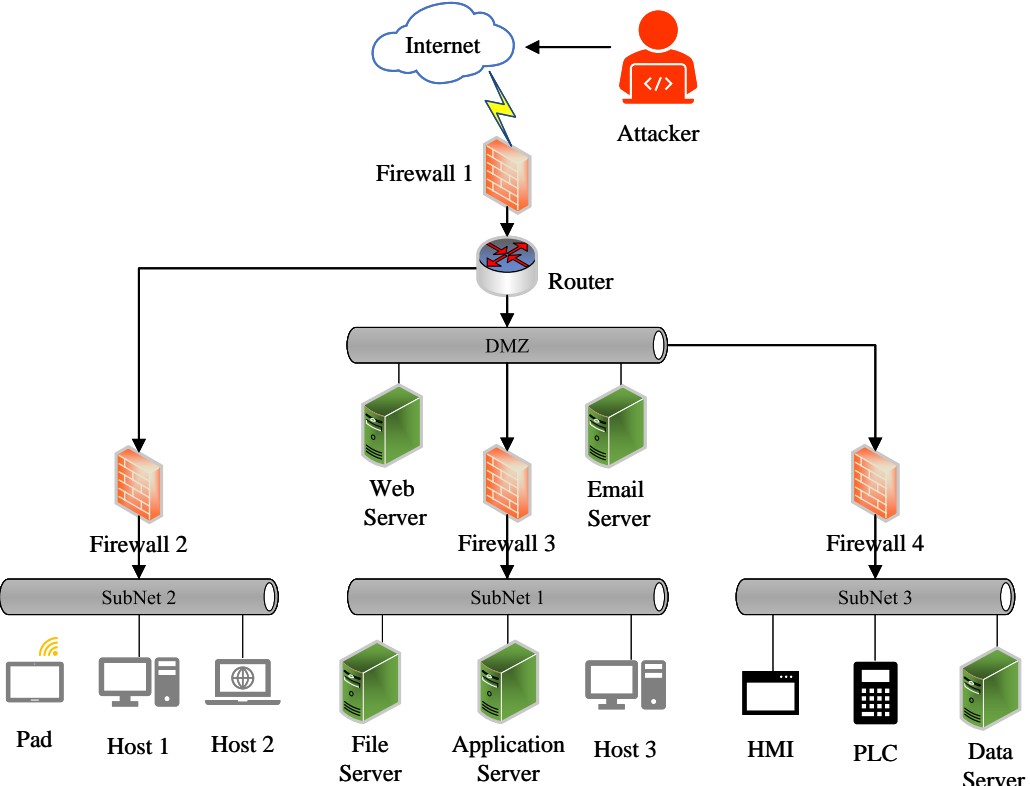

**Figure 5.** A general experiment network architecture.

*4.9. Two New Applications*

Thus far, this paper has focused on several dominating application scenarios. The following section will discuss some fresh applications, such as social engineering attack analysis, fake cyber threat intelligence identification, and so on.

### 4.9.1. Social Engineering

Social engineering, in short, is a sort of cyberattack where the attacker takes advantage of human vulnerability by engaging in social interactions to break cybersecurity [132]. Cyberspace security has been seriously compromised by social engineering. A social engineering ontology in cybersecurity is developed by Wang et al. [133] towards protecting social engineering cyberattacks, as well as a method for evaluating it by its applications. The ontology defines eleven essential entity concepts, as well as 22 types of relationships, which significantly comprise or impact the social engineering area. It offers a structured and explicit knowledge architecture for understanding, analyzing, reusing, and sharing social engineering field knowledge. The KG was also created using fifteen social engineering attack events and scenarios, and it was comprehensively assessed using seven application examples (in 6 analysis patterns) based on query methods.

### 4.9.2. Combating Fake Intelligence

Internet users today receive a considerable volume of fake cybersecurity intelligence. In order to get rid of this kind of information, Mitra et al. [134] built a system that captures provenance information and displays it together with the CTI captured. Together with enhancing the exiting CSKG model to combine intelligence provenance, this study fused provenance graphs with CSKG. The reasoning capabilities of CSKG enforce rules that aid in the preservation of reliable information while discarding the rest. Moreover, classes that capture provenance can be added to the CSKG schema, providing us with more information about the data's source. However, the details and datasets of this novel KG were not given.

Apart from that, Xiao et al. [135] developed a KG embedding method to predict software security entity within-type and across-type interactions. Analysts can expand their understanding of software security by discovering such missing connections between existing entities. However, this CSKG is not open-source, so we could not read the details of it. In addition, the mentioned white paper [117] reported several other application scenarios of CSKG technology as well as its two classical reasoning methods. Despite the fact that there was some limitation in stating the adequate details, the application scenarios, such as ATT&CK threat modeling, APT threat hunting, intelligent security operation, cyberspace surveying and mapping, supply chain security, and cyber-physical system protection, were outlined and forecasted by the white paper. There are two broad categories of reasoning technologies based on CSKG: relational reasoning based on graph representation learning and multi-relational reasoning methods based on neural networks.

## 5. Discussion and Research Opportunities

At the moment, some progress has been achieved in the use of KGs in a variety of knowledge-driven cybersecurity tasks, but it is still in its infancy. In this part, we first provide a quick overview of various methodologies to identify the gap, followed by proposing research opportunities in relevant aspects of CSKG.

**(1) Open-source dataset construction**

In reviewing the literature, no open-source data were found on the perfect solution to all the problems. For the task of CSKG construction, further research should be carried out on building a dataset that could cover all the dimensions, such as the knowledge data from MITRE, the CTI obtained from open sources, the environment data (e.g., assets, attributions, and their topology), and behavior data (such as network alerts, terminal alerts, and logs), as shown in Figure 6.

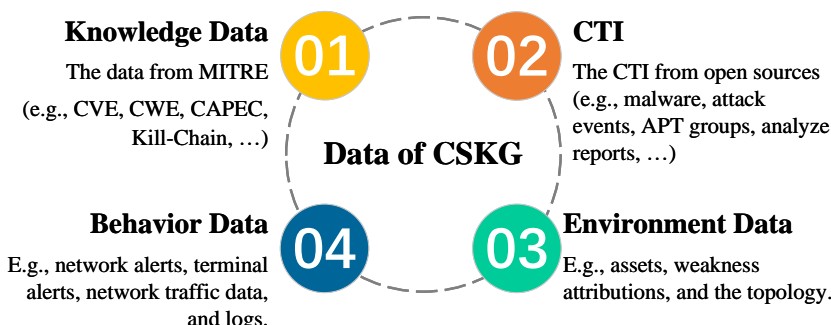

**Figure 6.** A general experiment network architecture.

The adequate annotated cybersecurity datasets are indispensable for training or validating the IE models, even for the pre-training language model or the prompt-based language models. However, existing datasets could not support this task well because of several drawbacks: first, most of them are designed for only one information extraction task (i.e., entity extraction) and rarely for two IE tasks; secondly, because of different self-designed ontologies and different research targets, the entity and relationship types are various; thirdly, the existing datasets are in a single language (i.e., English), which could not satisfy the requirement of multilanguage; finally, annotating the corpus manually remains the primary way of offering initial data for the model in the vertical domain.

Further research should be undertaken to investigate the new multilanguage cybersecurity IE dataset building based on comprehension and reliable data sources. This potential dataset should be annotated in a standard format and with a statement document. In the aspect of the annotated method, to lessen the reliance on the annotated vertical corpus, semi-supervised or unsupervised extraction approaches, as well as prompt-based generating methods, can be investigated.

**(2) The construction of a dynamic cybersecurity knowledge graph**

There are well-developed frameworks for knowledge graph building. To establish large-scale knowledge bases, both top-down [136] and bottom-up [137] building approaches can be utilized. In the field of cybersecurity, the former one is more popular (i.e., it designs a cybersecurity ontology schema first, then extracts the knowledge required by the schema from the corpus), which relies heavily on expert knowledge. The automatic ontology construction technology (also known as ontology learning) should still be considered necessary for the timely collection of emerging knowledge during the process of ontology update.

Conventional knowledge graphs mainly focus on the entities, their relations, attributions, etc., which are relatively deterministic and static knowledge. With the development of KG research and the demand for field applications, event knowledge and dynamic knowledge, such as temporal information, conditional relationships, causal information, and event subordination relationships, will inevitably be included. Considerably more work will need to be done to represent the cybersecurity event knowledge and support relevant logical reasoning by building a cybersecurity event temporal knowledge graph.

**(3) The application scenarios of the cybersecurity knowledge graph**

Although the construction technologies of CSKG are stable, there is still no unified open-source KG that is accepted by everyone. KGs, while their usefulness and utility are frequently incomplete, redundant, and ambiguous, can lead to uninformative query results. As a result of the different application demands of various scenarios, researchers have to rebuild a new knowledge graph every time. This survey has made a comprehensive review of the application scenarios of CSKG, but at present, the CSKG function proposed above mainly remains on the query and display functions provided by Neo4j. This does not fully exploit the KG's potential to automate reasoning. Therefore, it is still not clear how to use it to solve some practical problems in the cybersecurity domain. KG completion is just one of the many applications of knowledge reasoning technology. To achieve a new

understanding, additional exploration and studies based on reasoning technology should be conducted.

The semantic gap between CSKG and logs is the key to restricting the application of CSKG to attack path investigation. By supplementing relevant knowledge, this semantic gap can be filled, and the semantic association between CSKG and log can be realized. The most essential future task will be to improve the connection between the CSKG and internal knowledge of network, particularly the cyber-physical system, as well as to apply the KGs' automated reasoning ability and association analysis ability to uncover risks and network situational awareness.

**(4) The evaluation criterion of the cybersecurity knowledge graph**

CSKG has potential uses in both defensive and offensive scenarios across most cybersecurity activities, despite its youth. Currently, there are no established evaluation standards for the KG. The accuracy, precision, and F1 value are frequently used by researchers to assess the information extraction model. Hits@n, mean rank (MR), as well as mean reciprocal rank (MRR) could be used for assessing the triple prediction model's reasoning ability and use the query-based cases to demonstrate the KG's query and visualization capabilities. These are not ideal for a comprehensive analysis of a knowledge graph. For example, we cannot claim that the KG may be used in certain situations to demonstrate that it is superior to other KGs. Accordingly, future studies on proposing the evaluation standards for CSKG are, therefore, recommended.

## 6. Conclusions

In this review, we have provided a critical overview of the various works on the cybersecurity knowledge graph's application scenarios. To begin, this article provides a quick explanation of CSKG's origins, ideas, and building methods. Then, several open-source datasets that are available for building cybersecurity knowledge graphs and the information extraction task, and their drawbacks, are illustrated. In the fourth chapter of this paper, we carried out a comparative study of the many publications that expound on the most recent advances in the application scenarios of CSKG. A novel comprehensive classification framework was developed for describing the related works from 9 main aspects and 18 subclasses. Finally, new study options have been proposed based on a consideration of the inadequacies of the present research.

Cybersecurity teams could utilize CSKG to better intuitively understand threat intelligence, network posture, relationships, and attributes of security entities. CSKG could serve as a foundation for understanding the knowledge of cybersecurity, analyzing data of cybersecurity, and discovering the patterns of cyberattacks and abnormal features. It is hoped that this research will contribute to a deeper understanding of how to apply cybersecurity knowledge graphs in industrial practice.

**Author Contributions:** Conceptualization, Z.D. and Y.Z.; methodology, K.L. and F.W.; software, K.L.; validation, S.L. and Z.Y.; investigation, K.L. and F.W.; resources, K.L.; data curation, K.L.; writing—original draft preparation, K.L.; writing—review and editing, S.L. and Z.Y.; visualization, K.L. and F.W.; supervision, Z.D. and Y.Z.; project administration, Z.D.; funding acquisition, Y.Z. All authors have read and agreed to the published version of the manuscript.

**Funding:** This research was partially funded by The Science and Technology Innovation Program of Hunan Province, grant number 2021RC3076 and Training Program for Excellent Young Innovators of Changsha, grant number KQ2009009.

**Conflicts of Interest:** The authors declare no conflict of interest.

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
