# Peer review of "Recent Progress of Using Knowledge Graph for Cybersecurity"

_electronics, doi:10.3390/electronics11152287_

Round 1

Reviewer 1 Report

This research gives a quick overview of the cybersecurity knowledge graph’s core concepts, schema, and building methodologies. 

suggestions:

1. in "4.1.2. Security assessment, 4.2.2. Threat hunting, 4.2.3. Intrusion detection, 4.2. Threats Discovery and 4.3. Attack investigation" should have the summary tables better.

2. The research should list the process of how to collect the articles to review. Like: "The researcher applied a relevant set of keywords:????. These keywords are limited to the title, abstract and keywords search archives published between ??? and  ???. The database search returned a total of ??? publications." 

Reviewer 2 Report

The article is dedicated to an overview of the cybersecurity knowledge graph’s core concepts, schema, and building methodologies. Existing applications of this concept have proven their potential in countering cyber attackers.

The overview of the present article is a basis for clarifying the state of problems in the development of research in the field of CSKG as a specific Knowledge Grapf.

The presented classification system with linked 13
works from 9 core categories and 18 subcategories are useful.

Without going into details about the contents, I need to acknowledge the substantial contribution in the discussion section. The conclusion of this part is very significant. I consider that they are related to actual practice.

 A note:

It is unclear whether the authors " give a curated datasets collection" or review datasets...

Reviewer 3 Report

This research contains valuable information and adds to the science of knowledge. The authors discussed the research topic (knowledge graph for cybersecurity) comprehensively.  However, the authors should accurately address the below comments.

-          Keywords: We suggest that the authors should remove keywords such as “cyber security” and “knowledge graph;” because these keywords are already found in the review article title. It is better that they replace them with other keywords to increase the reach of the article.

-          Authors should improve the title "Other reasoning tasks".

-          Some paragraphs are very long and should be broken down into smaller paragraphs to be clearer such as page14-lines583-625 … etc.

-          English Writing: This article requires moderate proofreading. Author should check the entire article to remove all moderate drawbacks in terms of English writing.

-          References List: References are not arranged in the text. Some search names in the reference list begin an uppercase letter for each word (such as [3], [6] ... etc.) and others use only an uppercase letter in the first word (such as [1], [2] … etc.), authors should standardize style. Some references do not contain enough information such as references [17], [20], [54] … etc. Authors should proofread the entire list of references to make it mistakes-free.
